# Erosion of Stumble Correction Evoked with Superficial Peroneal Nerve Stimulation in Older Adults during Walking

**DOI:** 10.3390/jfmk9020094

**Published:** 2024-05-27

**Authors:** Ryan Brodie, Marc Klimstra, Drew Commandeur, Sandra Hundza

**Affiliations:** 1Motion and Mobility Laboratory, University of Victoria, Victoria, BC V8P 5C2, Canada; 2School of Exercise Science, Physical and Health Education, University of Victoria, Victoria, BC V8W 3P2, Canada; 3Canadian Sport Institute Pacific, Victoria, BC V9E 2C5, Canada; 4International Collaboration on Repair Discoveries (ICORD), Vancouver, BC V5Z 1M9, Canada

**Keywords:** aging, reflexes, stumble correction, peroneal nerve stimulation, falling, tripping, older adults

## Abstract

In healthy young adults, electrical stimulation of the superficial peroneal cutaneous nerve (SPn) innervating the dorsum of the foot has been shown to elicit functionally relevant reflexes during walking that are similar to those evoked by mechanical perturbation to the dorsum of the foot during walking and are referred to as stumble corrective (obstacle avoidance) responses. Though age-related differences in reflexes induced by mechanical perturbation have been studied, toe clearance has not been measured. Further, age-related differences in reflexes evoked by electrical stimulation of SPn have yet to be determined. Thus, the purpose of this study was to characterize age-related differences between healthy young adults and older adults with no history of falls in stumble correction responses evoked by electrical stimulation of the SPn at the ankle during walking. Toe clearance relative to the walking surface along with joint displacement and angular velocity at the ankle and knee and EMG of the tibialis anterior, medial gastrocnemius, biceps femoris and vastus lateralis were measured. The combined background and reflex toe clearance was reduced in the older adults compared with the young in mid-early swing (*p* = 0.011). These age-related differences likely increase fall risk in the older adult cohort. Further, age-related changes were seen in joint kinematics and EMG in older adults compared with the young such as decreased amplitude of the plantarflexion reflex in early swing in older adults (*p* < 0.05). These altered reflexes reflect the degradation of the stumble corrective response in older adults.

## 1. Introduction

Age-related differences in motor control and biomechanics have been shown to influence human gait and postural control [1,2]. However, some older adults are more prone to experiencing falls than others and it is currently unclear which age-related changes are part of normal healthy aging and which specifically contribute to fall risk. It is therefore important to understand alterations during walking in healthy older adults with no fall history compared with young adults to better understand the normal aging process such that it can be differentiated from the clinical changes that contribute to fall risk [3]. Studying the preservation of characteristic responses to perturbations during walking have been established as an effective means to evaluate the integrity of neural control [4,5,6], particularly given their important regulatory functions during human locomotion [7,8]. Characteristic stumble corrective (obstacle avoidance) responses are seen in response to perturbations to the dorsum of the foot during walking in young healthy adults. These stumble corrective strategies are similar with both mechanically and electrically evoked responses [9]. Electrical perturbations are delivered by the stimulation of the superficial peroneal nerve at the ankle innervating the dorsum of the foot, whereas mechanical perturbations involve mechanically tapping the dorsum of the foot. Characteristic obstacle avoidance responses in young healthy adults include reduced dorsiflexion (i.e., relative plantarflexion) and increased knee flexion during the swing phase compared with walking with no perturbation [8,10,11]. 

In response to mechanical perturbation to the dorsum of the foot during walking, older adults displayed reduced amplitude in the activity in some leg muscles and more failures to clear the obstacles compared with young adults; however, the joint kinematic responses were no different between the old and young adults [12]. The lack of significant kinematic differences between age groups could possibly be explained by the limited sensitivity of the kinematic measurement tool (i.e., electric goniometers). Though toe clearance is considered a key factor contributing to fall risk [13], it is currently not known if it is reduced after a mechanical perturbation to the dorsum of the foot during walking in older adults compared with young adults. Reflexes evoked by electrical stimulation of the tibial nerve innervating the sole of the foot have been compared between healthy young and older adults (i.e., those with no fall history) [14]; however, reflex responses to electrical stimulation of the superficial peroneal nerve (SPn) have not yet been investigated. Using electrical stimulation to evoke responses allows for more precise control over the perturbation location and intensity of the stimulus and activates a relatively larger portion of the cutaneous field than can be activated with mechanically evoked perturbations.

Understanding age-related degradation of the stumble correction response (i.e., SPn stimulation) in those individuals with no fall history provides insight into normal age-related decline in the sensorimotor system and is of particular importance given the putative functional importance of this reflex for trip and fall avoidance during walking. While there are multiple contributing factors to the stumble correction response, such as musculoskeletal, mechanical and physiological, quantifying the age-related decline in the neural control of the stumble correction response in older adults with no fall history can not only provide a baseline for comparison to those with a fall history but also has potential to provide a novel approach for establishing early neuromechanical markers in those individuals with a propensity for future fall risk. To this end, our research question was to understand what differences in reflex responses to electrical stimulation of the superficial peroneal nerve during walking (perturbed) exist between younger and older adults. This includes measures of toe clearance and joint kinematics and motor activity of muscles (electromyography) acting at the knee and ankle. We also wanted to compare background toe clearance, EMG and kinematics during walking without stimulation (unperturbed). We hypothesized that there would be diminished reflex responses and age-related differences in EMG, joint kinematics and toe clearance response in the old compared with the young. 

## 2. Materials and Methods

### 2.1. Participants

A total of 9 healthy older adults aged 70 and older (OLD; 5 male and 4 female, mean age 76.7 ± 4.8 years) and 10 younger adults (YOUNG; 5 male and 5 female, mean age 25.4 ± 5.4 years) free of any known neurological, musculoskeletal or metabolic impairment participated in this study. Subjects were screened using the Canadian PAR-Q+ questionnaire and medical clearance was requested for participants who answered Yes to any question. Inclusion criteria for this study required participants to be able to walk unassisted at least 200 m and to have a mini-mental state exam score of 24 or greater. Participants were excluded if they had a history of musculoskeletal or neurological disorders. Informed consent was obtained and this study was conducted in accordance with the University of Victoria Human Research Ethics Board.

### 2.2. Protocol 

Participants walked on a treadmill for approximately 9 min at a self-selected pace. Cutaneous reflexes were evoked during walking by stimulating the superficial peroneal nerve (SPn). Muscle activity and kinematics were measured during stimulated and non-stimulated walking cycles referred to as “perturbed” and “unperturbed”, respectively. 

### 2.3. Nerve Stimulation

Non-noxious percutaneous electrical stimulation was delivered to the SPn, innervating cutaneous receptors on the dorsum of the foot at the ankle. Stimulating electrodes were placed near the crease of the anterior surface ankle joint. Stimulation was generated by a Grass S88 stimulator (Grass Instruments, AstroNova, Brossard, QC, Canada) connected in series with an SIU5 isolator and a CCU1 constant current unit (Grass Instruments, AstroNova, Brossard, QC, Canada). Stimulation consisted of trains of 1 ms square-wave pulses (5 × 1 ms at 300 Hz) at approximately two times the intensity of the threshold for radiating paresthesia over the dorsum of the foot such that the participant reported clear radiating paresthesia towards the first toe (2.34 ± 0.7 and 2.33 ± 0.3 times for the old and young adults, respectively (*p* = 0.68)). Stimulation was delivered pseudorandomly throughout the gait cycle every 1 to 3 full gait cycles. Approximately 240 stimulus events were captured during each walking trial.

### 2.4. Electromyography 

The skin was lightly abraded and cleansed with alcohol, and disposable Ag-AgCL (model T3425, Thought Technology, Montreal, QC, Canada) surface EMG electrodes were applied in bipolar configuration longitudinal to the predicted path of the muscle fibers (2 cm interelectrode distance) over the tibialis anterior (TA), medial gastrocnemius, (MG), vastus lateralis (VL) and biceps femoris (BF) muscles ipsilateral to the site of nerve stimulation. A common ground electrode for the EMG was placed over electrically neutral tissue about the knee. Using a wired EMG system (model P511, Grass Instruments, AstroNova, Brossard, QC, Canada), signals were amplified and bandpass was filtered at 10–300 Hz. 

### 2.5. Kinematics and Gait Parameters 

Ankle and knee joint kinematics and toe clearance were recorded with an 8 camera Vicon T20S 3D motion analysis system (Vicon Motion Systems, Oxford, UK). Kinematic reconstruction of anatomical landmarks was based upon the 6 degrees of freedom model (Collins et al., 2009) [15], along with anatomical landmarks defined within Visual 3D v4.96.09 (C-Motion, Germantown, MD, USA). Joint angle and joint angular velocity (calculated as the time differential of joint angular displacement) and toe height data were collected. Toe clearance was determined as the vertical height of the lateral toe (5th metatarsal head) marker relative to the walking surface. 

### 2.6. Data Acquisition and Analysis

The approach to data acquisition and analysis was the same as previous work [14]. EMG data were sampled at 1000 Hz using a 16-bit A/D converter connected to a computer using custom-written LabView 2013 software (National Instruments, Austin, TX, USA). Kinematic and EMG data collection were synchronized. Kinematic data were sampled at a rate of 100 Hz using Vicon Nexus 1.7.2 software and analyzed using Visual 3D v4.96.09 software (C-Motion, Germantown, MD, USA). Kinematic data were interpolated for comparison with EMG data and Butterworth filtered at 10 Hz using custom-written Matlab 2013b (Mathworks, Natick, MA, USA) software. Post acquisition, the EMG and kinematic data were partitioned into 16 equal bins across the gait cycle, with bins 1–10 representing stance phase and 11–16 representing swing phase [4,8]. Sweeps of data for each sampling period contained 100 ms before the stimulus and 200 ms after (300 ms total per sweep). EMG and kinematic responses to nerve stimulation (perturbed) within each step cycle bin were each averaged. EMG and kinematics recorded without stimulation (unperturbed background) within each step cycle bin were also averaged. EMG and kinematics reflexes (subtracted traces) were calculated by subtracting the averaged unperturbed background data from the data with stimulation (perturbed) at each step cycle bin (10–20 observations per bin).

EMG data were full-wave rectified prior to averaging. Reflexes were only considered significant if they exceeded a 2 SD band calculated from pre-stimulus subtracted values. EMG traces were examined for the cumulative average over the 125 ms post-stimulus (ACRE125) (see [8]). Mean reflex and background EMG (bEMG) were normalized to the maximum average bEMG during the unperturbed walking cycle. Angular displacement reflexes were taken as the maximal excursion in mean subtracted traces within a window of 70–220 ms post-stimulus to reflect electromechanical delay in human skeletal muscle [16]. Angular velocity reflexes were taken as the peak amplitude of mean subtracted velocity within this window. Background angular displacement was taken as the maximal excursion in mean unperturbed traces within a window of 70–220 ms post-stimulus. Background angular velocity was taken as the peak amplitude of mean unperturbed velocity within this window.

### 2.7. Statistics

Descriptive statistics include mean, standard deviation (SD) and standard error of the mean (SEM). Using Statistica 10.0 (Tibco, Santa Clara, CA, USA), a 2 × 16 repeated measures analysis of variance (ANOVA) was conducted separately for background EMG and kinematics (joint angular displacement and velocity and toe clearance), and for EMG and kinematics reflexes (subtracted) to determine main effects for age and significant age–bin interactions. A repeated measures ANOVA accounts for within subject variability and allows for direct comparison between the OLD and YOUNG at each bin of the gait cycle. Tukey was used to investigate post hoc significant differences between age groups at each bin. Student’s *t*-test was used to compare stimulation intensities between cohorts. Significance was set at *p* < 0.05.

## 3. Results

### 3.1. Reflexes 

#### 3.1.1. Toe Clearance Reflexes 

Group averages for toe clearance reflex (subtracted) during perturbed walking for OLD and YOUNG are shown in Figure 1a. There was a significant main effect for age (F(1, 17) = 6.776, *p* = 0.019) and a significant age–bin interaction for toe clearance reflexes (F(15, 345) = 1.79, *p* = 0.035). Tukey post hoc analysis showed that the OLD had less toe clearance than the YOUNG, approaching significant difference at bin 12 (*p* = 0.061).

#### 3.1.2. Ankle Kinematic and EMG Reflexes

Group averages shown in Figure 2 highlight significant differences between age cohorts in ankle kinematics and related EMG reflex amplitudes. There was a significant main effect for age (F(1, 17) = 9.627, *p* = 0.006) and a significant age–bin interaction for ankle angular displacement reflexes (F(15, 345) = 2.242, *p* = 0.006). Post hoc analysis showed that there was reduced reflex plantarflexion displacement in the OLD compared with the YOUNG during swing in bin 10 (see Figure 2a). 

There was a significant main effect for age (F(1, 17) = 5.261, *p* = 0.035) and a significant age–bin interaction for ankle angular velocity reflexes (F(15, 345) = 3.886, *p* < 0.001). Post hoc analysis showed that there was reduced plantarflexion velocity at bin 10 (see Figure 2b). In TA, there was a significant main effect for age (F(1, 17) = 5.721, *p* = 0.029) and no significant age–bin interaction. In MG and SOL, there was no significant interaction or main effect. 

#### 3.1.3. Knee Kinematic and EMG Reflexes 

Group averages for knee kinematics and related EMG reflexes across the age cohorts are shown in Figure 3. A significant interaction was seen in both angular displacement (F(15, 255) = 2.25, *p* = 0.005) and angular velocity (F(15, 255) = 2.125, *p* = 0.009) knee reflexes. However, post hoc analysis showed no significant differences between OLD and YOUNG at any bins for angular displacement or angular velocity. No significant main effects or interactions were seen in VL or BF reflex amplitudes.

### 3.2. Background 

#### 3.2.1. Background Toe Clearance

Group averages for OLD and YOUNG for toe clearance throughout an unperturbed gait cycle are shown in Figure 1b. There was no main effect for age or age by bin interaction. 

#### 3.2.2. Background Ankle Kinematics and EMG

Group averages of background ankle angular displacement and angular velocity during unperturbed gait cycles are shown in Figure 4. There was a significant main effect for age (F(1, 17) = 8.549, *p* = 0.009) and a significant age–bin interaction (F(15, 255) = 6.226, *p* < 0.001) for ankle angular displacement. Post hoc analysis showed differences between OLD and YOUNG at bins 10, 11 and 12. The OLD had greater dorsiflexion displacement through the stance–swing transition at bin 10 and reduced plantarflexion displacement in early swing at bins 11 and 12 compared with the YOUNG (See Figure 4). 

There was a significant age–bin interaction for ankle angular velocity (F(15, 255) = 3.987, *p* < 0.001). Post hoc analysis showed that the OLD had reduced plantarflexion velocity at bin 10 and reduced dorsiflexion velocity at bin 13 compared with the YOUNG (Figure 4b). For TA, there was no significant main effect for age or age–bin interaction. In MG, there was no age–bin interaction, but there was a main effect for age (F(1, 17) = 17.30, *p* < 0.001), reflecting the greater level of overall muscle activity in the OLD compared with the YOUNG. 

#### 3.2.3. Background Knee Kinematics and EMG

Group averages of joint angular displacement and angular velocity for the knee during unperturbed gait cycles are shown in Figure 5. For background knee angular displacement, there was a significant age–bin interaction (F(15, 255) = 3.082, *p* < 0.001). Post hoc analysis showed no specific bin-related differences between OLD and YOUNG. For background knee angular velocity, there was a significant age–bin interaction (F(15, 255) = 3.024, *p* < 0.001). Post hoc analysis revealed specific bin-related differences between OLD and YOUNG at bin 15, where the OLD had reduced extension velocity. 

For VL there was a significant age–bin interaction (F(15, 255) = 2.42, *p* = 0.003). Post hoc analysis showed no specific bin differences between age cohorts. For BF, there was a significant age–bin interaction (F(15, 255) = 4.858, *p* < 0.001). Post hoc analysis showed that BF activity was higher in the OLD than the YOUNG during stance bins 3–4.

### 3.3. Combined Reflexes and Background

Group averages for OLD and YOUNG for toe clearance throughout gait cycle during walking with perturbations are shown in Figure 1c. These toe clearance values reflect the sum of the differences between OLD and YOUNG for background and reflex toe clearance. A main effect for bin (F(15, 255) = 182.40 *p* < 0.001) and a significant interaction (F(15, 255) = 2.096, *p* = 0.011) were found for toe clearance during walking with perturbations. In the OLD, toe heights were significantly closer to the ground during mid-swing at bin 12. A main effect for age between the cohorts (F(1, 17) = 10.23, *p* = 0.005) and a significant interaction (F(15, 255) = 7.675, *p* < 0.001) were seen for ankle displacement during walking with perturbations. Post hoc analysis showed differences between age cohorts at bin 10, 11 and 12. A significant main effect for bin (F(15, 255) = 408.35, *p* < 0.001) and a significant age–bin interaction (F(15, 255) = 3.38, *p* < 0.001) were also found for knee displacement. Post hoc analysis found no specific bin-related differences between cohorts.

## 4. Discussion

This work demonstrates significant changes in toe clearance and ankle joint kinematics during mid-swing in OLD compared with YOUNG in response to reflexes evoked by electrical stimulation of the cutaneous afferents on the dorsum of the foot. This work also demonstrated age-related altered gait profiles during unperturbed walking. Blunted stumble correction reflexes in healthy older adults superimposed on altered age-related gait profiles may result in decreased capacity to avoid contact with obstacles while walking, potentially leading to increased risk of tripping. Thus, this reflects early degradation of neuromechanical control and the functional capacity of stumble correction during gait in healthy older adults. 

### 4.1. Toe Clearance

The current results are the first to demonstrate a significant reduction in toe clearance mid swing (bin 12) in the OLD compared with the YOUNG, which is seen when the toe clearance reflex (approaching significance) is superimposed on the toe clearance data from unperturbed gait profiles in these healthy older adults. This results in a greater risk of the foot not clearing the obstacle. This reduced amplitude in the toe clearance reflex in the OLD compared with YOUNG is in keeping with previous findings by Schillings and colleagues (2005), who demonstrated that older adults had more failures to clear the obstacles during walking compared with young adults [12]. Our previous work [14] also demonstrated reduced toe clearance in healthy older compared with younger adults, with perturbation to the sole of the foot. Barrett and colleagues found toe clearance to be a relevant factor in differentiating fallers from non-fallers [17]; therefore, the differences in toe clearance seen in the current study could reflect an increased risk for tripping in the OLD compared with the YOUNG, despite the fact that this older cohort have not yet experienced a fall. Thus, the current results demonstrate early degradation of neuromechanical control and reduced stumble correction capacity during gait in healthy older adults. This specific biomarker could reflect the normal aging process, an accumulation of lifestyle choices or the presence of preclinical comorbidities. Regardless of the cause, this biomarker may suggest that these older adults are at increased risk of falls compared with young adults despite no prior history of falls. 

### 4.2. Ankle

Previous studies in healthy young adults using electrical stimulation of the SP nerve at the ankle [8,18] and mechanical perturbation to the top of the foot [10,11] consistently demonstrated a stumble corrective reflex characterized by plantarflexion at the ankle in early swing supported by TA suppression and increased knee flexion in mid-swing supported by BF excitation. The increased knee flexion brings the foot away from the ground to allow the ankle to plantarflex to avoid the obstacle while having the foot clear the obstacle. Although the general pattern of the stumble corrective response was observed in both age cohorts in the current study, in the OLD cohort there was an erosion of the stumble corrective response at the ankle and in combination with the background gait profile there was a significant change in the ankle kinematics. 

At the ankle, older subjects displayed reduced reflex amplitude in plantarflexion displacement and velocity during early-mid swing (i.e., main effect age and significant reduced amplitude at bin 12). Though this reduced amplitude in plantarflexion reflex would not have contributed to the reduced toe clearance seen in the OLD, this blunted reflex could contribute to decreasing the effectiveness of obstacle avoidance during swing phase of gait (i.e., eroded stumble correction response). This reduced plantarflexion occurred in the OLD in spite of their increased amplitude in the inhibitory reflex in TA dorsiflexor motor output. This reduced plantarflexion response in the older adults could result from increased ankle joint stiffness [19] or, alternatively, reflect age-related differential strength losses in ankle musculature and the functional responses to stimulation. The strength capacity of plantarflexors degrades during normal ageing while the strength of dorsiflexors remains relatively maintained [20]. The increased amplitude in the inhibitory reflex in TA, therefore, may be required to mitigate the reduced contribution of plantarflexors in maintaining joint stiffness as the foot passes the obstacle. 

During unperturbed walking cycles (background), the current results show some important differences in the kinematics and muscle activity across gait cycle between the age cohorts that are consistent with other research. Similar to Winter and colleagues [21], we found reduced ankle range of motion (i.e., main effect age and significant reduction bin 10–12) in the OLD and reduced velocities (i.e., main effect age and significant reduction bin 10 and 13) compared with the YOUNG. Such reductions at distal joints have been previously shown to be linked with a redistribution of joint torques favoring proximal musculature [22]. The greater dorsiflexion seen in the OLD at late stance may be linked with reduced push-off leading into early swing [21] and reduced plantarflexion displacement seen throughout swing. The reduced ankle joint kinematics seen in the OLD may be related to a more “tentative” gait pattern with age [23]. During swing, background muscle activity in MG was greater in the OLD compared with the YOUNG, despite there being reduced plantarflexion in the OLD. This may relate to plantarflexor strength degradation seen with normal ageing [20] or age-related increase in passive tissue resistance at the ankle [19].

In the OLD cohort, the erosion of the stumble corrective response at the ankle in combination with an altered background gait profile at the ankle resulted in a significant reduction in plantarflexion at the ankle (i.e., main effect age and significant reduction bin 10–12), which may result in a decreased ability in older adults to avoid foot contact with the obstacles. The reduced plantarflexion in swing in the OLD for both background and reflex ankle kinematics would encourage increased toe clearance for the OLD compared with the YOUNG and therefore did not account for the relative reduced toe clearance seen in the OLD in the current study. It is possible that the reduced ankle plantarflexion reflex represented compensation in an attempt to keep the foot further from the ground. Alternatively, perhaps there was a decreased momentum of plantarflexion associated with the reduced plantarflexion velocity in both the reflex and background (unperturbed) gait profile, which may in turn have reduced the magnitude of the plantarflexion displacement reflex during walking in the OLD. 

### 4.3. Knee

There was a significant age–bin interaction for the knee ROM of reflex with a pattern of reduced knee flexion in early-mid swing in the OLD, though the finding was not significant at a specific bin. Similarly, Schillings and colleagues (2005) found that older participants displayed reduced knee flexion in early swing compared with the young; however, this finding was also not significant. This lack of significance could be due to the studies being underpowered for detecting change in this variable. The knee flexion angular velocity reflexes also showed a significant age–bin interaction, with no significance for a specific bin; however, there was a pattern of lower amplitude in the OLD during mid-swing and greater in late swing. The lack of significant reflexes in EMG in BF and VL was in contrast to previous work that found reduced amplitude in the BF reflex for late latency responses [12]. This contrast may reflect the different methodologies employed. The current study averaged the reflex amplitude over 50–125 ms, whereas the previous work reflected peak reflexes in late latency (~100–150 ms).

There was a significant age–bin interaction for the background knee ROM and, though the finding was again not significant at a specific bin, there was a pattern of reduced knee flexion in early to mid-swing in the OLD. The knee flexion angular velocity during unperturbed walking also showed a significant age–bin interaction with a significantly greater velocity into flexion in the OLD during late swing (i.e., significance at bin 15). It is possible that the increased knee velocity flexion in the background profile in late swing may contribute to delayed foot placement during the lowering strategy (late swing), which is associated with falling in older adults [24,25]. The lack of significant knee reflex and background results do not explain the significant reduction in mid-swing toe clearance. It is likely that the combined alterations to both background and reflex kinematics at the trunk, hip, knee and ankle result in reduced toe clearance, with the foot acting as the end-point effector for the lower limb kinematic chain, with the overall differences reflecting a general degradation in neural control. 

### 4.4. Clinical Implications

Blunted reflexes to SP nerve stimulation in healthy older adults superimposed on altered background kinematic and EMG gait profiles resulted in significant reduction in toe clearance mid-swing in OLD compared with YOUNG. These changes reflect early degradation of neuromechanical control and the functional capacity to respond to perturbations during walking in older adults. This knowledge adds to a basic understanding of age-related changes in neural function. Further research comparing reflex control in older adults with and without a fall history is required to determine whether this early neuromechanical degradation reflects normal age-related changes or the prodromal signs of fall risk. If this early degradation reflects an early biomarker of increased fall risk, it suggests the potential use of cutaneous reflexes in quantifying degradation of neuromuscular control and its contribution to fall risk. Evaluating the neuromechanical obstacle avoidance response through cutaneous reflex analysis may allow early identification of those at future risk of falls and enable timely presentation of appropriate prevention strategies. Additionally, the restoration of the normal stumble corrective response and unperturbed gait properties in the old could be the goal of fall prevention interventions.

### 4.5. Limitations and Future Directions 

The current study did not record EMG from all muscles, nor was there measurement of kinematics in all joints (e.g., hip) that may contribute to the toe clearance and joint kinematic results seen here. Further, the current study did not measure other potential contributing factors to toe clearance and joint kinematics such as joint and muscle stiffness, muscle strength, etc. For example, it has been noted that reduced lower limb strength is associated with fall risk [26]. Lastly, this research was conducted while walking on a treadmill, reducing the generalizability to overground walking [27]. Future research could include all factors (physiological, neural, mechanical) as covariates or independent factors to determine specific contributions to the end kinematic result. 

## Figures and Tables

**Figure 1 jfmk-09-00094-f001:**
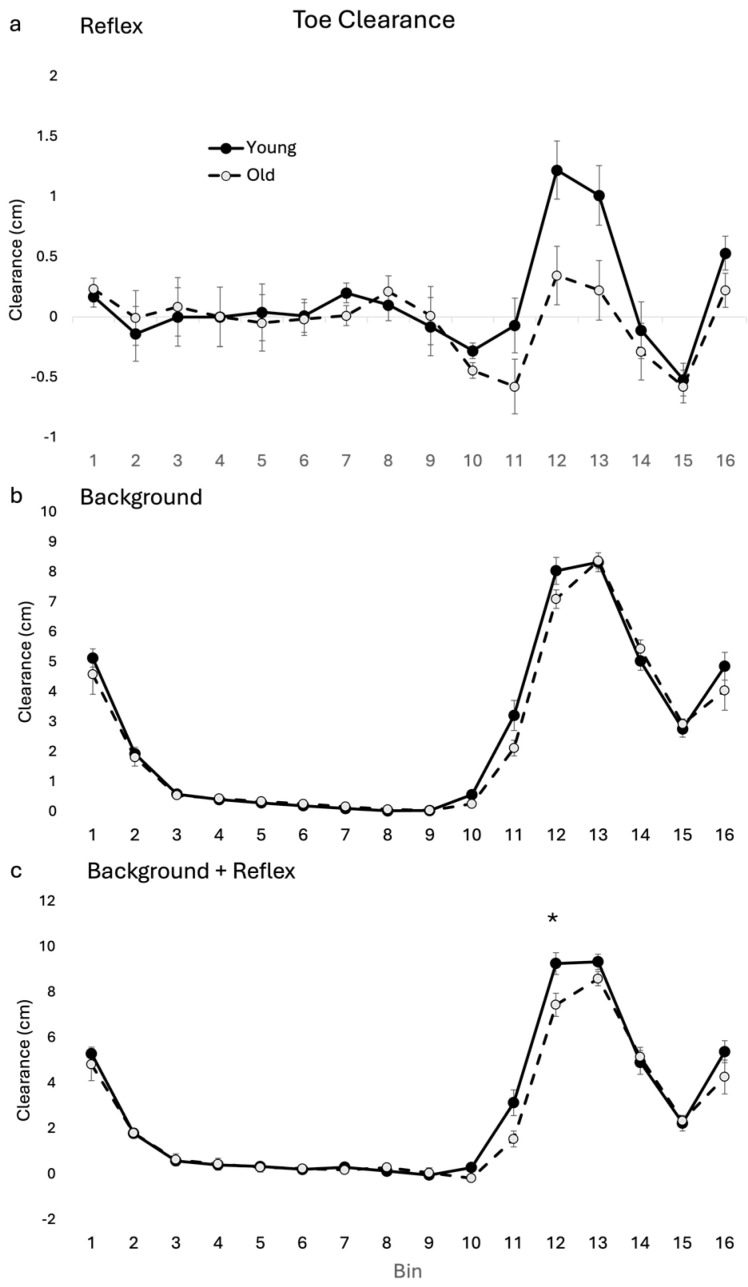
Toe clearance group means for OLD and YOUNG across the gait cycle for (**a**) reflex (subtracted (perturbed minus unperturbed), (**b**) background (unperturbed) and (**c**) background + reflex (subtracted plus unperturbed). * indicates significant difference between group difference at that bin. *p* < 0.05 level.

**Figure 2 jfmk-09-00094-f002:**
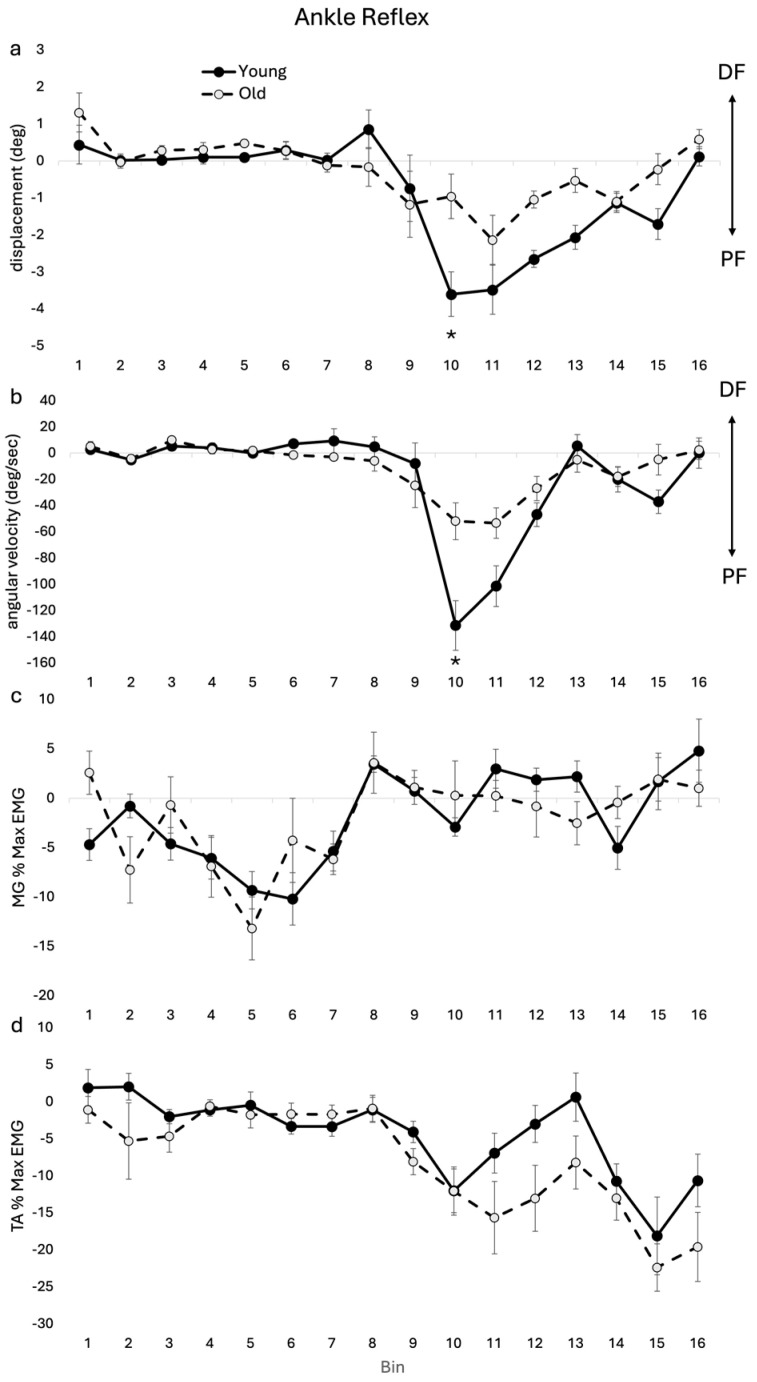
Ankle reflex (subtracted (perturbed minus unperturbed)) group means for kinematics and EMG for OLD and YOUNG across the gait cycle for (**a**) displacement (deg), (**b**) angular velocity (deg/sec), (**c**) MG %max EMG, and (**d**) TA %max EMG. EMG values are represented as the average cumulative reflex EMG after 125 ms (ACRE125) and are normalized to peak background muscle activation throughout the gait cycle for each subject. * indicates significant difference between group difference at that bin. *p* < 0.05 level.

**Figure 3 jfmk-09-00094-f003:**
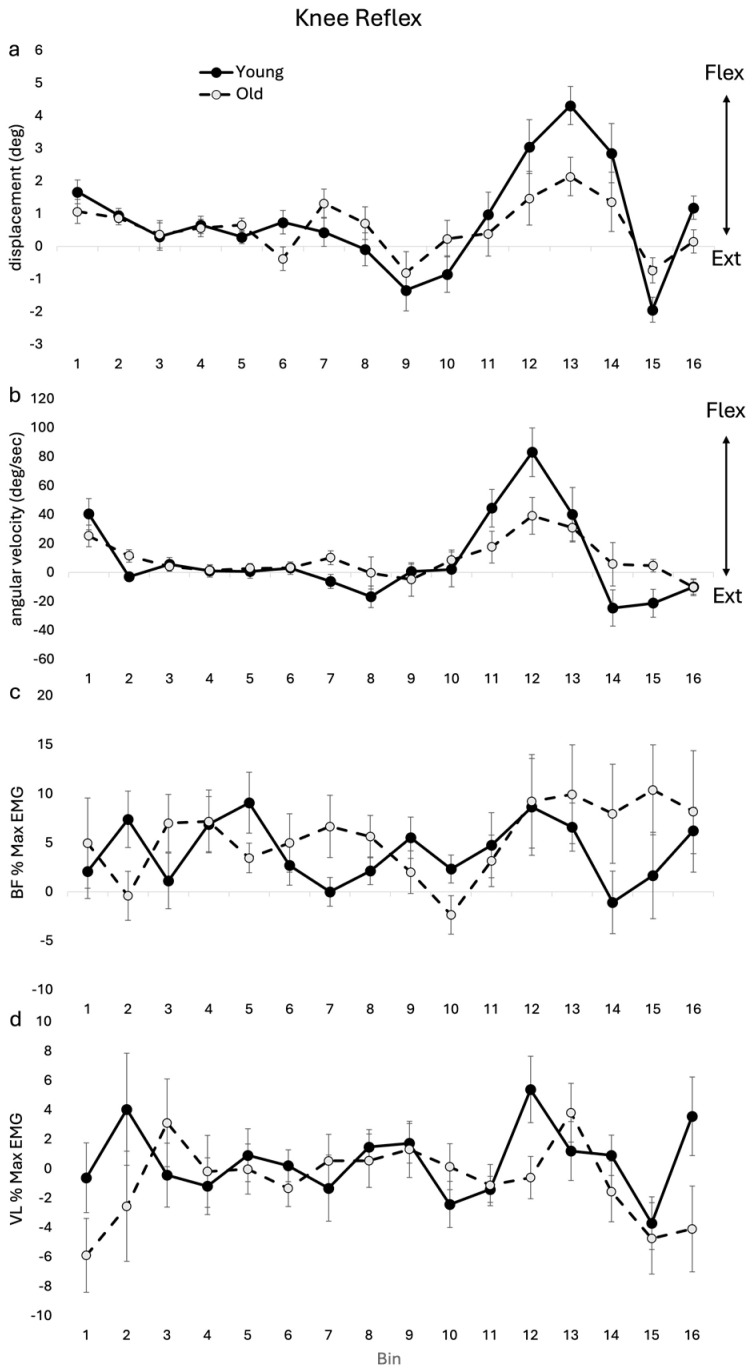
Knee reflex (subtracted (perturbed minus unperturbed)) group means for kinematics and EMG for OLD and YOUNG across the gait cycle for (**a**) displacement (deg), (**b**) angular velocity (deg/sec), (**c**) BF %max EMG, and (**d**) VL %max EMG. EMG values are represented as the average cumulative reflex EMG after 125 ms (ACRE125) and are normalized to peak background muscle activation throughout the gait cycle for each subject.

**Figure 4 jfmk-09-00094-f004:**
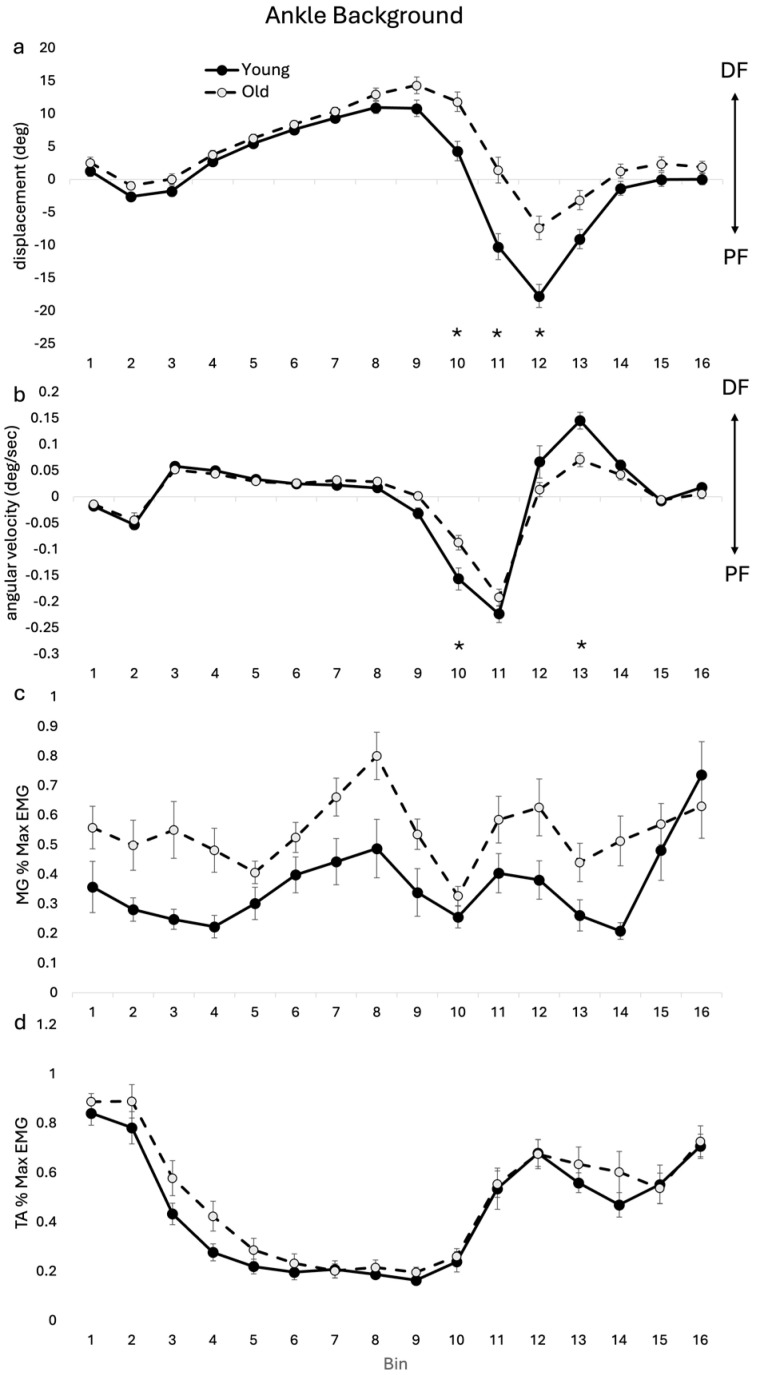
Ankle background (unperturbed) group means for kinematic and EMG for OLD and YOUNG across the gait cycle for (**a**) displacement (deg), (**b**) angular velocity (deg/sec), (**c**) MG %max EMG, and (**d**) TA %max EMG. EMG values are represented as the average cumulative reflex EMG after 125 ms (ACRE125) and are normalized to peak background muscle activation throughout the gait cycle for each subject. * indicates significant difference between group difference at that bin. *p* < 0.05 level.

**Figure 5 jfmk-09-00094-f005:**
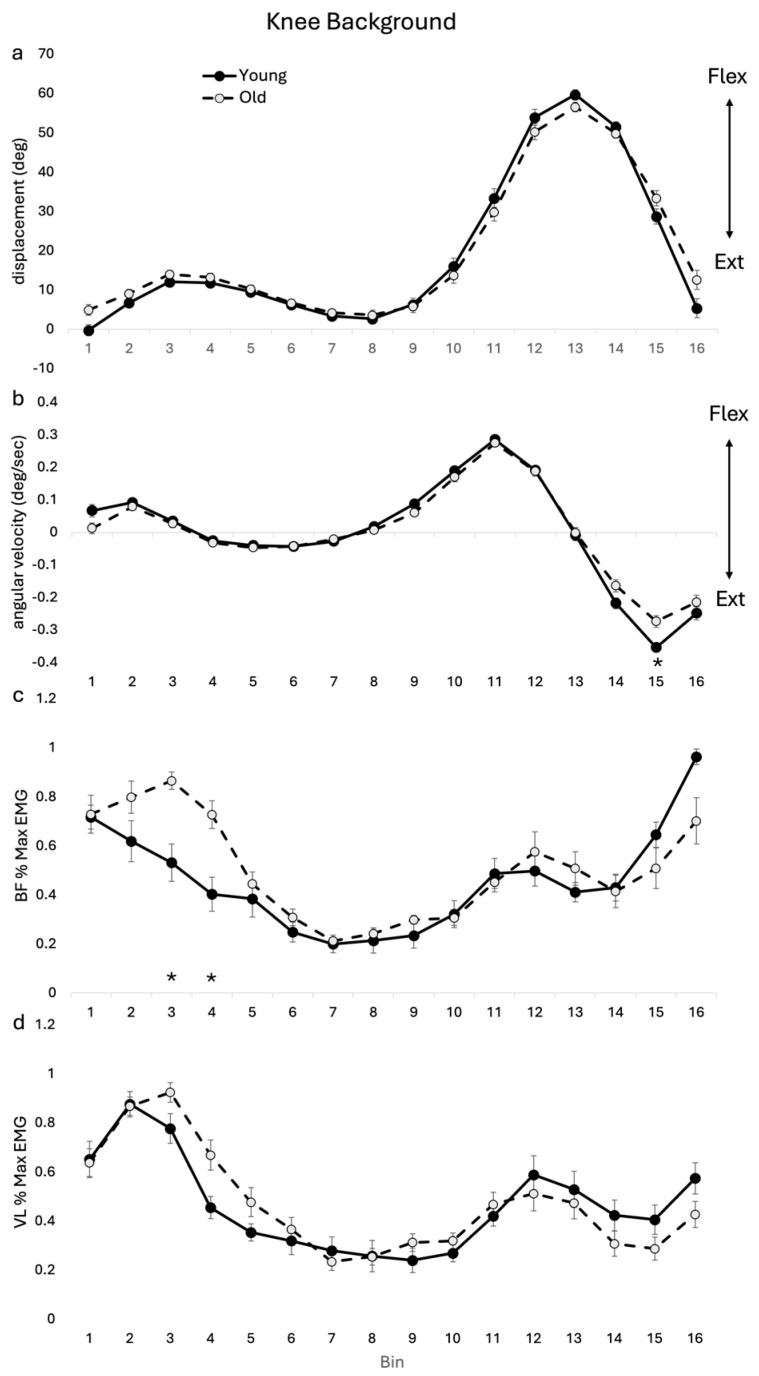
Knee background (unperturbed) group means for kinematic and EMG for OLD and YOUNG across the gait cycle for (**a**) displacement (deg), (**b**) angular velocity (deg/sec), (**c**) BF %max EMG, and (**d**) VL %max EMG. EMG values are represented as the average cumulative reflex EMG after 125 ms (ACRE125) and are normalized to peak background muscle activation throughout the gait cycle for each subject. * indicates significant difference between group difference at that bin. *p* < 0.05 level.

## Data Availability

Data used in this study are not able to be shared outside of the original research group.

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
