# Peer review of "Erosion of Stumble Correction Evoked with Superficial Peroneal Nerve Stimulation in Older Adults during Walking"

_jfmk, 2024, doi:10.3390/jfmk9020094_

Round 1
Reviewer 1 Report
Comments and Suggestions for Authors
Thank you for submitting your work. Here are some of the comments that can improve your manuscript:
1) Figures: Please include x axis in each of the figures denoted as a,b,c,d.
2) I would suggest summarizing your primary findings in starting of discussion and then talking about details in subheadings.
Author Response
Thank you for submitting your work. Here are some of the comments that can improve your manuscript:
- Figures: Please include x axis in each of the figures denoted as a,b,c,d.
We thank the reviewer for this suggestion and have added the axis. As it is the same axis for all of the figures we have only added the numbers and not the title “Bin” to reduce visual clutter.
- I would suggest summarizing your primary findings in starting of discussion and then talking about details in subheadings.
We thank the reviewer for this comment and have clarified the summary of finding at the beginning of the discussion.
Reviewer 2 Report
Comments and Suggestions for Authors
Suggestions in the file attached

Author Response
27: I recommend that it should be noted that this fall risk is the one due to the aspects spoken
above, as there are another reasons for risk fall
We thank the reviewer for this comment and have made changes to the abstract
67: I would suggest to write “those individuals” or similar instead of those alone
We have made this change.
80: I think it would be interesting to be a bit more concrete, as the choice of the elderly
population, if we look at its age, has to be carefully done, as it is quite possible that some
issues during their life could affect nerve functioning.
We have added more information on the inclusion/exclusion criteria for the subjects. It is also important to note that the stimulation intensity for the old and young participants are similar.
(2.34 ± 0.7 and 2.33 ± 0.3 times for the old and young adults respectively (p = 0.68)
99,100,101: I recommend to clarify the reason for the “pseudorandomly” delivered stimulus
and the events captured. (maybe references or technical reasons).
The stimulation was delivered in a somewhat random (i.e pseudorandom) nature so that the participants were less able to predict or anticipate. The exact time the stimulation would occur. Pseudorandom in this context meant that a stimulation pulse occurred once within 1-3 cycle timeframe. In this way the stimulation was not completely random.
107: It would be interesting, if possible, to have some image of the setting for more clarity.
Unfortunately, we do not have a picture of the setup and we have now recently changed labs and equipment.
161: Unless there´s a technical reason for it I suggest that parts of every figure should be a bit
more separated. I think that it would be easier to check for the reader
We thank the reviewer for this suggestion and we have added an x-axis to every figure and separated them for greater clarity.
278: I propose that it should be noted that there might be more aspects concerning the
prodromal stage (which is an interpretation) as there can be more factors that affect the nerve
others than age (sedentary habits, etc.). I believe that it would be interesting to at least point it out
We thank the reviewer and have included your point that neuromechanical changes could result from normal aging, lifestyle choices or preclinical comorbidities.
Reviewer 3 Report
Comments and Suggestions for Authors
1. The introduction broadly sets out the purpose of the study. It would benefit from a more detailed articulation of the main hypotheses and specific research questions, clearly indicating the expected direction of the study's outcomes.
2. The statistical analysis section requires additional details on the use of multivariate analysis and the rationale behind the choice of statistical models. In particular, a deeper discussion on how the chosen statistical model impacts the study's results and the rationale for using specific models to verify interactions between age and control groups would be beneficial.
3. The experimental setup section should describe more systematically the measures taken to ensure consistency in the location and intensity of electrical stimulation at the ankle.
4. While the results section adequately explains differences in muscle responses by age, it lacks a detailed explanation of how these differences contribute to increased fall risk.
5. How can the study distinguish whether the age-related differences observed in responses are due to changes in the nervous system or physiological changes in the muscles?
Author Response
- The introduction broadly sets out the purpose of the study. It would benefit from a more detailed articulation of the main hypotheses and specific research questions, clearly indicating the expected direction of the study's outcomes.
We thank the reviewer and have added a more detailed articulation of the research questions and hypothesis.
- The statistical analysis section requires additional details on the use of multivariate analysis and the rationale behind the choice of statistical models. In particular, a deeper discussion on how the chosen statistical model impacts the study's results and the rationale for using specific models to verify interactions between age and control groups would be beneficial.
We thank the reviewer for this point. We have provided a rationale for the statistical model.
- The experimental setup section should describe more systematically the measures taken to ensure consistency in the location and intensity of electrical stimulation at the ankle.
We thank the reviewer for this point. We have provided greater explanations of the measures taken to ensure consistency in the location and intensity of stimulation. Stimulating electrodes were placed on or near the crease of the anterior surface ankle joint. Stimulation intensity was set approximately 2 x threshold for perception of paresthesia at the ankle such that stimulation at 2 x this threshold resulted in paresthesia radiating over the dorsum of the foot and towards the first toe. We confirmed with a t-test that the intensity of was not statistically significant between cohorts.
- While the results section adequately explains differences in muscle responses by age, it lacks a detailed explanation of how these differences contribute to increased fall risk.
We thank the reviewer for this comment. The EMG results in the current study do not clearly relate to fall risk in the way the age-related differences in toe clearance and joint kinematics do. This could result from us not measuring EMG from all muscles that may produce the toe clearance and joint kinematics. Further, we did not measure other potential contributing factors such joint and muscle stiffness, muscle strength, etc. We have described this in a new Limitations section in discussion. Further we did not measure kinematics or EMG from other potentially contributing joints such as the hip.
- How can the study distinguish whether the age-related differences observed in responses are due to changes in the nervous system or physiological changes in the muscles?
We thank the reviewer for this comment. This is related to the point above and a limitation in measurement design. Future research could include all factors (physiological, neural, mechanical) as covariates or independent factors to determine specific contributions to the end kinematic result. Regardless of the cause we still see the differences between the old and young in this cohort.
Reviewer 4 Report
Comments and Suggestions for Authors
This study characterized age-related differences between healthy young adults and older adults with no history of falls in stumble correction responses evoked by electrical stimulation of the superficial peroneal cutaneous nerve at the ankle during walking. Although the study is interesting, there are some concerns/suggestions that authors may need to address before considering the manuscript suitable.
1. Pg1; Ln 23-25. Elaborate more on the results subsection of the abstract. Consider supporting your results with statistical indicators.
2. Pg1; Ln 26-27. “Together these age-related differences likely represent the prodromal sign of increased fall risk”. Does the data of this study support this claim?
3. The age-related fall is not only related to physiological changes in the nervous system, but changes in other systems like the musculoskeletal system can also contribute to falling. This should be clarified somewhere in the introduction.
4. Pg 2; Ln 31-65. Could authors provide further context and background information about the topic in light of previous studies?
5. Pg 2; Ln 66-78. I appreciate the author for the clear and compelling justification of the study. They articulated the importance and relevance of the research question, which helped to establish a solid foundation for the study and its intended outcomes. However, I would like to see the initial hypothesis statement by the end of the introduction to set the stage for the subsequent results and discussion sections, providing a roadmap for the study direction.
6. I have a major concern regarding the study power. The study is likely not powered enough for the design and outcome measures. Data from just 9 healthy adults and 10 young adults are not sufficient to yield definitive conclusions.
7. No mention of priori sample size estimations. On what basis did you decide an entire sample size of 19 participants?
8. Pg 2-3; Ln 92-101. Please provide a detailed description of the percutaneous electrical stimulation so that other researchers can replicate it. Where were electrodes placed? why did you use intensity two times the intensity of the threshold for radiating paresthesia over the dorsum of the foot?
9. Pg 3; Ln 98. Consider adding the measurement unit for the employed stimulation intensity.
10. Pg 3; Ln 102-106. Which wireless EMG instrument did you use? Provide a synopsis of its specifications and recording parameters.
11. It would be interesting if the description of percutaneous electrical stimulation and EMG recording were supported by a picture illustrating the application and electrode placement.
12. Pg 3; Ln 112-113. “Joint angle and joint angular velocity (calculated as the time differential of joint angular displacement)”. Please, provide proper credit to the source of this information.
13. Results are clear and well-presented, and I can follow based on the study objective.
14. The discussion reads well. However, it can benefit from further comparison with previous results.
15. I would also recommend dedicating a space in the discussion section to acknowledge the study limitations and their potential impact on the application of data. Additionally, providing directions for future research would help identify areas that require further investigation or offer suggestions to improve the study design.
Author Response
- Pg1; Ln 23-25. Elaborate more on the results subsection of the abstract. Consider supporting your results with statistical indicators.
We thank the reviewer for this point. We have added statistical indicators to our results in the abstract.
- Pg1; Ln 26-27. “Together these age-related differences likely represent the prodromal sign of increased fall risk”. Does the data of this study support this claim?
We thank the reviewer for this point. We have modified the language in the abstract as this statement may be too interpretative.
- The age-related fall is not only related to physiological changes in the nervous system, but changes in other systems like the musculoskeletal system can also contribute to falling. This should be clarified somewhere in the introduction.
We thank the reviewer for this point. We have added in the introduction that indeed musculoskeletal changes can contribute to fall risk.
- Pg 2; Ln 31-65. Could authors provide further context and background information about the topic in light of previous studies?
We thank the reviewer for this point. We believe we have already included all available relevant context from all available studies related to age related changes in reflex control.
- Pg 2; Ln 66-78. I appreciate the author for the clear and compelling justification of the study. They articulated the importance and relevance of the research question, which helped to establish a solid foundation for the study and its intended outcomes. However, I would like to see the initial hypothesis statement by the end of the introduction to set the stage for the subsequent results and discussion sections, providing a roadmap for the study direction.
We thank the reviewer for this point. We have added a hypothesis based on previous studies.
- I have a major concern regarding the study power. The study is likely not powered enough for the design and outcome measures. Data from just 9 healthy adults and 10 young adults are not sufficient to yield definitive conclusions.
We thank the reviewer for this point. We performed power calculations using previous research stimulating the tibial nerve for differences seen in toe clearance between old and young subjects. These calculations demonstrated that we needed a sample size of 4 per group for toe clearance.
- No mention of priori sample size estimations. On what basis did you decide an entire sample size of 19 participants?
We thank the reviewer for this comment. As described above we performed power calculations based on previous research that determined a minimum of 4 subjects per group. While we collected data on a larger number of participants in both cohort there were issues with data collection (i..e missing data due to marker malfunction) that minimized the cohort size to 9 and 10 for old and young respectively.
- Pg 2-3; Ln 92-101. Please provide a detailed description of the percutaneous electrical stimulation so that other researchers can replicate it. Where were electrodes placed? why did you use intensity two times the intensity of the threshold for radiating paresthesia over the dorsum of the foot?
We thank the reviewer for this comment. We have improved the detail of the nerve stimulation methodology.
- Pg 3; Ln 98. Consider adding the measurement unit for the employed stimulation intensity.
We thank the reviewer for this comment. The measurement unit is relative to the perception of the participant and is included.
- Pg 3; Ln 102-106. Which wireless EMG instrument did you use? Provide a synopsis of its specifications and recording parameters.
We thank the reviewer for this comment. We have added more detail in this section. We used a wired P511, Grass Instruments EMG system (AstroMed Inc.) not a wireless system.
- It would be interesting if the description of percutaneous electrical stimulation and EMG recording were supported by a picture illustrating the application and electrode placement.
We thank the reviewer for this comment. We believe with the increased detail in the description of electrode placement for stimulation and EMG recording that an image would not significantly add to the paper.
- Pg 3; Ln 112-113. “Joint angle and joint angular velocity (calculated as the time differential of joint angular displacement)”. Please, provide proper credit to the source of this information.
We thank the reviewer for this comment. Angular velocity is by definition the time-derivative of angular displacement.
- Results are clear and well-presented, and I can follow based on the study objective.
We thank the reviewer for this comment.
- The discussion reads well. However, it can benefit from further comparison with previous results.
We thank the reviewer for this comment. There are limited directly relevant cutaneous reflex studies comparing older and young adults during walking. We have however added reference to previous work as they relate to the added section on Limitations and future directions.
- I would also recommend dedicating a space in the discussion section to acknowledge the study limitations and their potential impact on the application of data. Additionally, providing directions for future research would help identify areas that require further investigation or offer suggestions to improve the study design.
We thank the reviewer for this comment. We have added a section on Limitations and future directions which significantly adds to the paper.
Round 2
Reviewer 4 Report
Comments and Suggestions for Authors
The authors addressed and clarified most of the comments/suggestions raised in the first round. However, I am not convinced by their response to the issue of the study power.
The authors claimed that they had performed power calculations using previous research stimulating the tibial nerve for differences seen in toe clearance between old and young subjects. These calculations demonstrated that they needed a sample size of 4 per group for toe clearance.
Since no data was presented input settings used for the sample size calculation, I cannot be confident of the statistical power.
Author Response
We thank the reviewer for their care in consideration of this manuscript.
With respect to the power/sample size calculations here are the details for your consideration.
In a previous study that we performed:
https://www.ncbi.nlm.nih.gov/pmc/articles/PMC5966742/
We found a difference in toe clearance of 9.5cm in the young compared to 7.8 cm for the old at bin 12 +_ 0.7 cm.
From this we performed power calculations:
In the present study we have approximately the same differences at bin 12 which would suggest an appropriate sample size for this current investigation.